# Eco-Physiological Adaptations of the Xylotrophic Basidiomycetes Fungi to CO_2_ and O_2_ Mode in the Woody Habitat

**DOI:** 10.3390/jof8121296

**Published:** 2022-12-13

**Authors:** Victor A. Mukhin, Daria K. Diyarova

**Affiliations:** Institute of Plant and Animal Ecology, Ural Division of the Russian Academy of Sciences, 620144 Yekaterinburg, Russia

**Keywords:** Basidiomycota, xylotrophic fungi, wood, gas mode, adaptations

## Abstract

The aim of this research is to study of eco-physiological adaptations of xylotrophic fungi (Basidiomycota, Agaricomycetes) to hypoxia, anoxia and hypercapnia as the main environmental factors that determine the activity of fungi in woody habitat. The study was carried out on seven species of polypore fungi widespread in the preforest-steppe pine-birch forests of the Central Urals, including both white (*D. tricolor*, *D. septentrionalis*, *F. fomentarius*, *H. rutilans*, *T. biforme*) and brown (*F. betulina*, *F. pinicola*) rot. Their CO_2_ and O_2_ gas exchange were analyzed in natural samples of woody substrates (*Betula pendula*, *Pinus sylvestris*) and basidiocarps by the chamber method using a CO_2_/O_2_ gas analyzer. It was shown that the intensity of O_2_ gas exchange is positively related to the oxygen concentration but is not very sensitive to a decrease in its content in the woody habitat. Xylotrophic fungi are able to completely exhaust the O_2_ in the habitat, and this process is linear, indicating that they do not have threshold values for oxygen content. Oxygen consumption is accompanied by an adequate linear increase in CO_2_ concentration up to 18–19%. At a concentration of 5–10%, carbon dioxide does not affect the gas exchange of xylotrophic fungi and can even enhance it, but at 20% it significantly reduces its intensity. Xylotrophic fungi are resistant to high CO_2_ concentrations and remain viable at 100% CO_2_ concentration and are capable of growth under these conditions. In an oxygen-free habitat, anaerobic CO_2_ emissions are recorded; when O_2_ appears, its consumption is restored to the level preceding anoxia. Xylotrophic fungi are the specialized group of saprotrophic microaerophilic and capnophilic facultative anaerobes adapted to develop at low oxygen and high carbon dioxide concentration, anoxia.

## 1. Introduction

Xylotrophic fungi (Basidiomycota, Agaricomycetes) are a group of organisms with relatively little biological diversity: 900 (former USSR), and 1700 (North America) species [1,2]. Most of them (57–75%) belong to Aphyllophoroid Hymenomycetes, and the smaller part (23–37%)—to Agaricoid ones. The minor component (2–6%) is Heterobasidioid fungi. According to Parmasto [3], they originated possibly from primitive unspecialized soil Basidiomycetes in the Cretaceous or earlier. Phylogenetic analysis based on the reconstruction of the origin of AA2 genes shows that white rot fungi could appear about 300 million years ago, at the end of the Carboniferous [4].

Currently, these are the only known organisms capable of the biochemical conversion of the lignocellulose complex and decomposition of wood without the participation of other decomposers [2,5,6,7,8]. Wood decomposition is an alternative to the photosynthesis process of the oxidative conversion of organic carbon from wood into CO_2_ and, given that forests are the largest terrestrial carbon reservoirs [9,10], xylotrophic fungi are not only globally significant emitters of carbon dioxide but also of corresponding scale consumers of oxygen [11].

Wood for them is not only a trophic resource, but also habitat, which in many of its parameters is very specific and, one might even say, extreme. One of its important features is low gas permeability and, as a result, for wood the special gas mode is typical. The main features of this gas mode are low (1.2–4.5%) concentrations of O_2_ and high (7.2–26.3—CO_2_ [12,13]. According to Hintikka and Korhonen [14], in coniferous wood decayed by xylotrophic fungi, the content of CO_2_ is 35 (1.6%), and in hardwood—88 (3.5%) times higher than in air. In aspen wood affected by *Phellinus tremulae*, at the initial stage of decomposition, the concentration of O_2_ and CO_2_ is 0.1–0.8% and 12.8–13.6%, respectively, at a more advanced stage, the content of O_2_ increases to 8.4–17.8%, while CO_2_ remains at the same level: 3.6–13.5%. At the final stage of decomposition, oxygen is 19.4–20.9%, and CO_2_ is 0.3–1.8%. Usually, in wood decomposed by fungi, O_2_ is about 6%, and CO_2_—is 14% [13].

If hypoxia and hypercapnia are obligate characteristics of the gas mode of the woody habitat, then oxygen-free conditions, anoxia, is most likely an episodic phenomenon that occurs under certain environmental conditions, primarily at high humidity. In wood that has low permeability to gases (spruce, larch), anaerobic conditions are formed at 50–70% humidity, and in easily permeable wood (birch, pine, poplar), at 100–110% [13]. The content of O_2_ in the wood also depends on temperature, increased temperature increases the intensity of its consumption by fungi and, as a result, the oxygen concentration in wood decreases, while CO_2_ increases [6]. Xylotrophic fungi are able to fully use the oxygen in the habitat and thereby create physiologically determined anoxia [15].

Xylotrophic fungi are adapted to the gaseous conditions of the woody habitat. This is indicated by the fact that they are (a) not sensitive to low oxygen concentrations (growth stops at an O_2_ concentration of 0.4%), (b) resistant to high concentrations of carbon dioxide (they grow at a CO_2_ concentration of 30–40%, and a 10% concentration stimulates growth), (c) remain viable in a long time stay viable in absence of oxygen in the woody habitat [7,13,14,15,16,17,18,19,20]. 

The gas mode, as we believe, is the main factor of the woody habitat, the presence of adaptations which is a necessary condition for the xylotrophic lifestyle of fungi. According to our hypothesis, these adaptations are microaerophilia, capnophilia, and facultative anaerobiosis, which form a complex of interrelated and interdependent adaptations of xylotrophic fungi to the gas mode of the woody habitat. 

The present work is devoted to the substantiation of this hypothesis.

## 2. Materials and Methods

### 2.1. Objects of Study

The study was carried out on the seven species: *Daedaleopsis tricolor* (Bull.) Bondartsev & Singer, *D. septentrionalis* (P. Karst.) Niemelä, *Fomes fomentarius* (L.) Fr., *Fomitopsis betulina* (Bull.) B. K. Cui, M. L. Han and Y. C. Dai, *F. pinicola* (Sw.) P. Karst., *Hapalopilus rutilans* (Pers.) Murrill, *Trichaptum biforme* (Fr.) Ryvarden. These are widespread polypore fungi [21], both white (*D. tricolor*, *D. septentrionalis*, *F. fomentarius*, *H. rutilans*, *T. biforme*) and brown (*F. betulina*, *F. pinicola*) rot (Figure 1). Woody debris of *Betula pendula* Roth and *Pinus sylvestris* L. with basidiocarps of these fungi were collected in preforest-steppe pine-birch forests of the Central Urals. Fungi were identified based on the morphological features of basidiocarps [21], and species names were checked by the Index Fungorum database [22]. In the laboratory, wood was cleaned of leaves and needles, the basidiocarps were separated from them and sawn into samples, the size of which, depending on specific tasks, varied from 2.6 to 7.5 cm in length and from 1.7 to 5.8 cm in diameter. 

The gas exchange of fungal basidiocarps and samples of wood decayed by them (hereinafter, substrates) was analyzed separately. This makes it possible to compare the carbon-oxygen gas exchange in two parts of the fungal organism developing in the air (basidiocarps) and wood (substrate mycelium). 

### 2.2. Assessment of the Intensity of Gas Exchange

Samples of basidiocarps and substrates were placed in sealed exposure chambers, the volume of which, depending on the aim of the experiment and the size of the samples, varied from 0.27 to 1.65 L. The concentration O_2_ and CO_2_ in the chambers were measured by CO_2_/O_2_ gas analyzer (“Microsensornaya Tekhnika”, Russia). This is a device combining the infrared and electrochemical principle of operation, equipped with an automated flow sampling system with a built-in micro-computer, with a measurement error ± 0.2 vol. %. Gas measurements were carried out either in automatic mode with an interval of 10–60 min (dynamics of O_2_ consumption and CO_2_ accumulation) or in manual mode (most of the experiments). The intensity of O_2_ and CO_2_ gas exchange was estimated from the difference between their initial and final concentration in the chambers in vol. %/h according to the formula: Consumption O_2_/emission CO_2_ = ΔO_2_/CO_2_/E × 60 × 273/T, 
where ΔO_2_/CO_2_—the amount of O_2_ consumed/CO_2_ emitted (%), E—the exposure time (min), T—the air temperature (K). 

### 2.3. Preparation of Gaseous Media with Different Contents of O_2_ and CO_2_

For the preparation of gaseous media with different oxygen content, 0.5-L chambers with samples of basidiocarps, substrates were filled with nitrogen of high purity (volume fraction 99.995%), and then high-purity O_2_ (volume fraction 99.95%) was injected with a syringe at a concentration of 5, 10, and 20% by volume. The actual content of O_2_ in the chambers was estimated using a gas analyzer, and it was close to the calculated one: 5.18 ± 0.12, 10.03 ± 0.14, and 20.17 ± 0.25%. Gaseous media with different CO_2_ contents were prepared similarly; 5, 10, and 20% high-purity CO_2_ (volume fraction 99.99%) was injected into 0.5 L chambers filled with air with a syringe at concentrations of 5, 10, and 20%. The actual CO_2_ concentration in the chambers, estimated using a gas analyzer, was close to the calculated one: 5.04 ± 0.20, 10.53 ± 0.14, and 20.20 ± 0.17%. The intensity of gas exchange depending on the oxygen concentration was estimated from the emission of CO_2_, and carbon dioxide—from the consumption of O_2_. Both high-purity nitrogen and high-purity carbon dioxide were used to create anoxic conditions. To do this, gases were pumped into the chambers until in them the gas analyzer registered zero O_2_. The absence of oxygen in the chambers was also monitored during subsequent measurements, and in order to eliminate or at least reduce the effect of carbon dioxide contained in the samples on the results even before they were placed in oxygen-free conditions, the nitrogen in the chambers was periodically renewed while maintaining anoxia. In an oxygen-free nitrogen medium, fungal gas exchange was assessed by the intensity of CO_2_ emission, and the viability of basidiocarp and substrate samples after their exposure to anoxic conditions was assessed by the intensity of O_2_ and CO_2_ gas exchange after pumping into the chamber air. 

### 2.4. Evaluation of Mycelium Growth under Anoxic Conditions

The assessment of the possibility and intensity of growth of xylotrophic fungi in the absence of O_2_ was performed on dikaryotic cultures of *D. septentrionalis* isolated from the basidiocarps of this fungus using the standard method [23]. Wort (4%)—agar (2%) was used as a nutrient medium for the isolation and preservation of cultures, as well as during experimental work. Petri dishes 10 cm in diameter with dikaryotic mycelium *D. septentrionalis* were placed closed in 0.5 L exposure chambers filled with nitrogen (*n* = 3) and carbon dioxide (*n* = 3). The control was three Petri dishes placed in chambers filled with air. All chambers were placed in a thermostat at a temperature of 20 ± 2 °C, and after 3 days the radial size of the mycelium was measured in four directions and the growth rate was calculated in mm/day. Then the chambers with dishes were filled with air and placed in a thermostat with the same temperature, after 3 days the radial growth of mycelium over this period was measured and the rate of growth in the air was calculated. 

### 2.5. Statistical Analysis

Statistical analysis was performed using the Statistica 8.0 program. Arithmetic means (*m*) are given with standard errors (*SE*), *t*-test or one-way analysis of variance (*ANOVA*) was used for multiple comparisons. The Spearman correlation coefficient (*R*) was used to characterize the relationships between variables. When describing the results of statistical evaluation, the values of the corresponding criterion and the levels of its significance are given. Statistical significance was indicated with a *p*-value < 0.05

## 3. Results

Gas exchange of xylotrophic fungi is typically aerobic, including consumption of O_2_ and emission of CO_2_ (Figure 2). The correlation coefficients indicate a close, functional level of connection between these processes, both in the gas exchange of basidiocarps (*R* = −0.99) and substrates (*R* = −0.99). An important feature of the carbon-oxygen gas exchange of basidiocarps and substrates is their ability to completely deplete O_2_. Zero oxygen content was recorded in the chambers with basidiocarps and *F. fomentarius* substrates, while the CO_2_ concentration in the chambers reached 18%. In chambers with *F. pinicola* basidiocarps, the minimum O_2_ concentration was 0.01%, and the maximum CO_2_ concentration was 19%. The substrates of *F. betulina* are characterized by a low activity of gas exchange, and over 2 days of their exposure, the O_2_ concentration in the chambers decreased from 20.7% to 1.7%, and the CO_2_ concentration increased to 15%. Both processes—consumption of O_2_ and release of CO_2_—with a high degree of reliability, both in the case of gas exchange of basidiocarps (R^2^ = 0.95–0.98) and substrates (R^2^ = 0.97–0.98) are described by linear regression equations (Figure 2). They show that the intensity of O_2_ consumption by basidiocarps and substrates is 13–20% higher than the CO_2_ emission. Correspondingly, the ratio of CO_2_ and O_2_ volumes, or the respiratory coefficient, is less than unity: in the basidiocarps *F. fomentarius* and *F. pinicola* it is 0.8–0.9 and in the substrates *F. fomentarius* and *F. betulina*—0.9. The quotient varies within 0.8–1.0 in the range of O_2_ concentrations from 19 to 0.01%, and CO_2_ from 0.8 to 19% and does not show a correlation with the content of these gases in the chambers.

In contrast to the respiratory coefficient, the O_2_ consumption by xylotrophic fungi closely and positively correlates with its concentration. This is typical both for the gas exchange of the basidiocarps *F. fomentarius* (R = 0.83, *p* = 0.01, n = 8), *F. pinicola* (R = 0.98, *p* = 0.0001, n = 13), and the substrates of *F. fomentarius* (R = 0.74, *p* = 0.003, n = 13) and *F. betulina* (R = 0.98, *p* = 0.0001, n = 17). At the same time, the intensity of O_2_ consumption is weakly dependent on O_2_ concentration (Figure 3). Thus, with a decrease in O_2_ from 17 to 0.15% (113 times), the intensity of its consumption by basidiocarps of *F. fomentarius* decreases only 1.2 times: from 3.21 to 2.58 vol. %/h. The same for *F. pinicola* basidiocarps: under decrease in O2 in chambers from 18 to 0.03% (600 times) O_2_ consumption decreases 1.5 times: from 2.42 to 1.60 vol. %/h. The intensity of O_2_ gas exchange of substrates reacts similarly to the oxygen concentration: in *F. fomentarius* it decreases by 1.5 times with a decrease in O_2_ from 19.5% to 0.25% (78 times), and in *F. betulina* by 1.7 times with a decrease in O_2_ by 12 times—from 19.7% to 1.6%. On average, as linear regression equations show, the intensity of O_2_ consumption decreases 20 (basidiocarps)—50–150 times (substrates) slower than its concentration in chambers (Figure 3).

Consumption of O_2_ by fungi is accompanied by an increase in CO_2_ concentration and, as the data show, the intensity of CO_2_ emission by basidiocarps closely but negatively correlates with its concentration: *F. fomentarius*—R = −0.97 (*p* = 0.001, n = 8), *F. pinicola*—R = −0.91 (*p* = 0.0001, n = 13). In the case of *F. betulina* substrates, a close negative relationship between the CO_2_ content in the chambers and the intensity of its emission is also recorded (R = −0.97, *p* = 0.0001, n = 17), while it is not observed in *F. fomentarius* substrates: R = −0.31, *p* = 0.28, n = 13. As in the case of O_2_ gas exchange, there is very little reaction of the intensity of CO_2_ emission to CO_2_ increase: decreases by 10–30 (basidiocarps)—40–130 times (substrates) more slowly than increase CO_2_ (Figure 4). 

The weak sensitivity of xylotrophic fungi to O_2_ concentration is confirmed by the data in Table 1 and Table 2. They show that in a nitrogen medium that does not contain CO_2_, but with different concentrations of O_2_ (5, 10 and 20%), the ratio of the volumes of emitted CO_2_ and consumed O_2_ fluctuates within 0.7–1.1 and does not show a relationship with the oxygen content (Table 1). Under these conditions also no relation between the concentration of O_2_ in chambers and the intensity of basidiocarps and substrates gas exchange (Table 2). 

The weak sensitivity of xylotrophic fungi to CO_2_ is shown by the data on the assessment of their gas exchange in chambers with an O_2_ concentration equal to its content in air, but with 0.04, 5, 10, and 20% CO_2_ (Table 3 and Table 4). According to them, at 5–10% CO_2_ in the chambers, the respiratory quotient is the same as at 0.04%, and the intensity of O_2_ gas exchange does not decrease, but even increases. However, at 20% CO_2_ concentration, a clearly pronounced negative effect on gas exchange is observed: the CO_2_/O_2_ ratio increases to 2.0–3.8 and gas exchange intensity decreases by two to six times compared with one at 5–10%.

The high resistance of xylotrophic fungi to carbon dioxide is evidenced by the fact that they remain viable even when they are kept in chambers with 100% CO_2_ concentration. Thus, after 72 h of exposure to such conditions, *D. tricolor* basidiocarps have the same O_2_ consumption intensity in the air as the initial one before exposure to a carbon dioxide medium. Similarly, such high concentrations of CO_2_ react and the gas exchange of *D. tricolor* substrates and the intensity of O_2_ consumption does not differ before and after exposure to 100% carbon dioxide (Table 5).

At 100% CO_2_ concentration, as our data show, xylotrophic fungi are capable of growth. Thus, in 100% carbon dioxide the dikaryotic mycelium of *D. septentrionalis* grows at a rate only two times less than in the air and when in chambers where CO_2_ is replaced by air, it grows at the same rate as in CO_2_ (Figure 5). 

The ability of *D. septentrionalis* to grow in a 100% carbon dioxide medium indicates not only its resistance to high concentrations of CO_2_ but also to the possibility of development in anaerobic conditions. This is confirmed by the data showing that the mycelium of *D. septentrionalis* grows in an anoxic nitrogen medium at the same rate as in air (Table 6).

In oxygen-free nitrogen media CO_2_ emission was also recorded both in basidiocarps and substrates, although at a lower intensity than in air (Table 7). Basidiocarps and substrate exposition in oxygen-free nitrogen media do not affect gas exchange in the air or lead to its decrease (Table 8).

## 4. Discussion

In air, the gas exchange of basidiocarps and substrates of xylotrophic fungi is typically aerobic: O_2_ and CO_2_ fluxes are functionally related, and multidirectional; their ratio 0.7–1.0 is in the range characteristic for aerobic respiration. At the same time, the aerobic carbon-oxygen gas exchange of xylotrophic fungi has a number of fundamental features: 1. The respiratory quotient does not show a connection with O_2_ in a wide range of concentrations—almost to zero; 2. The intensity of O_2_ consumption by xylotrophic fungi positively correlates with its concentration in its habitat, but very weakly reacts to O_2_ decrease: decreases by 10–30 (basidiocarps)—40–130 times (substrates) more slowly than O_2_ go down; 3. Xylotrophic fungi are able to fully use O_2_ and this process is linear and it is indicated that they do not have any critical, threshold values for the oxygen content. All this, in our opinion, indicates that xylotrophic fungi belong to microaerophiles, organisms adapted to develop in environments with low oxygen content [24].

Under conditions of hindered diffusion, a decrease in oxygen in the habitat is accompanied by an adequate increase in carbon dioxide. As in the case of O_2_ consumption, the intensity of CO_2_ emission reacts weakly to an increase in CO_2_ decreases 10–30 (basidiocarps)—40–130 times (substrates) slower than the CO_2_ content increase. At a 5–10% concentration, CO_2_ either does not affect the gas exchange of xylotrophic fungi, or even enhances it, but at 20% significantly (up to six times) reduces O2 consumption intensity. Xylotrophic fungi are very resistant to carbon dioxide, they remain viable in a 100% CO_2_ atmosphere and are capable to grow in these conditions. These are all undoubted signs of capnophiles—extremophilic organisms that successfully develop at 10–15% of CO_2_ concentration [25]. Resistance to CO_2_, or capnophilia, is most likely what fundamentally distinguishes xylotrophic fungi from litter saprotrophs, whose growth decreases in proportion to an increase in carbon dioxide concentration [14]. 

The fact that xylotrophic fungi are facultative anaerobes—organisms that use oxygen but are able also to do without it [25], is clearly evidenced by the following: they are able to remain viable in oxygen-free media for a long time and conduct CO_2_ gas exchange as well as growth. When O_2_ appears in the habitat, its consumption by fungi is restored and often at the pre-anoxic level. 

Thus, xylotrophic fungi there are extremophilic aerobic organisms, capable of development under low oxygen and prolonged anoxia as well as high carbon dioxide concentration, thanks to they have eco-physiological adaptations include microaerophilia, capnophilia and facultative anaerobiosis. These are three interrelated and interdependent adaptations of xylotrophic fungi to the gas mode of the woody habitat and if some of them are absent it reduces the adaptive significance of the others and makes it impossible for the successful development of fungi in wood. 

The gas mode of the woody habitat created by themselves, on the one hand, gives them competitive advantages over other groups of xylobiont organisms, and, on the other hand, is a selection factor for the accompanying organisms. So, low oxygen content and its absence contribute to the development of anaerobic bacteria in wood. Indirectly, this is indicated by methane emissions from decomposed xylotrophic fungi wood [26,27,28,29,30].

## Figures and Tables

**Figure 1 jof-08-01296-f001:**
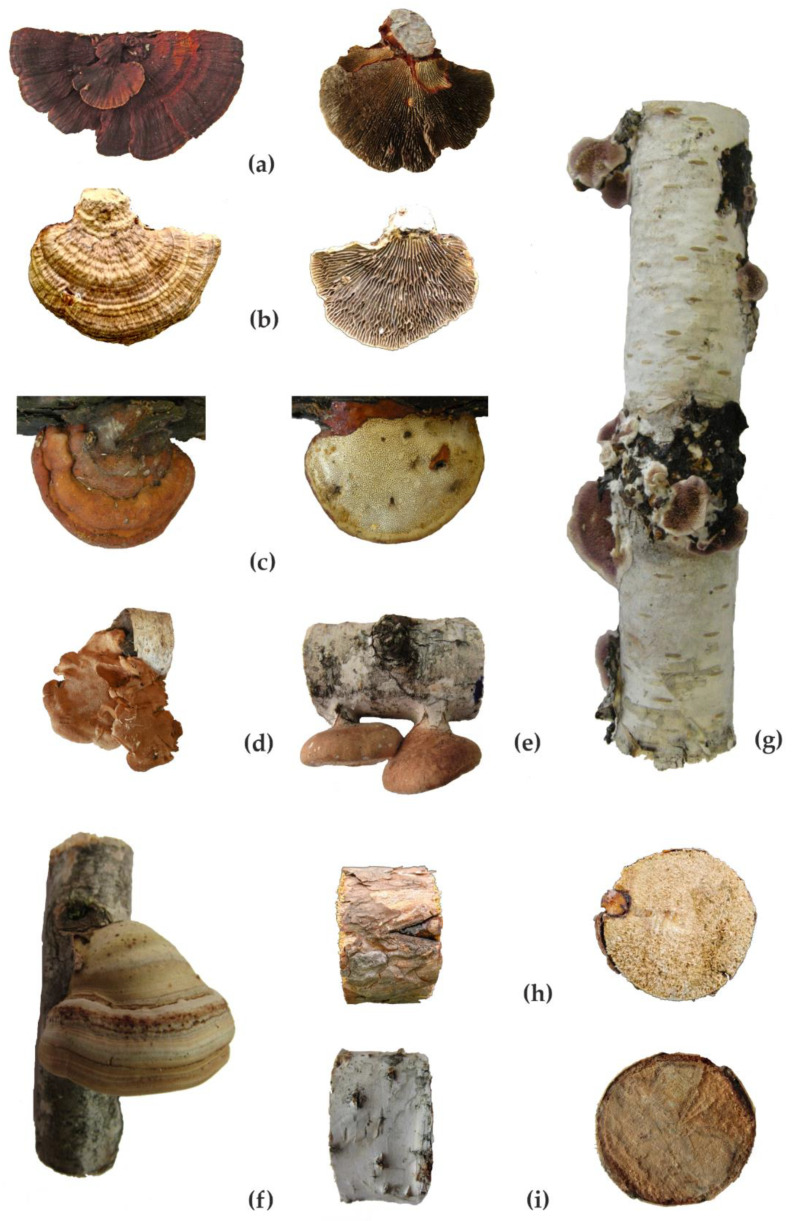
*Daedaleopsis tricolor* (**a**), *D. septentrionalis* (**b**), *Fomitopsis pinicola* (**c**), *Hapalopilus rutilans* (**d**), *F. betulina* (**e**), *Fomes fomentarius* (**f**), *Trichaptum pargamenum* (**g**); substrates: *Pinus sylvestris* (**h**), *Betula pendula* (**i**).

**Figure 2 jof-08-01296-f002:**
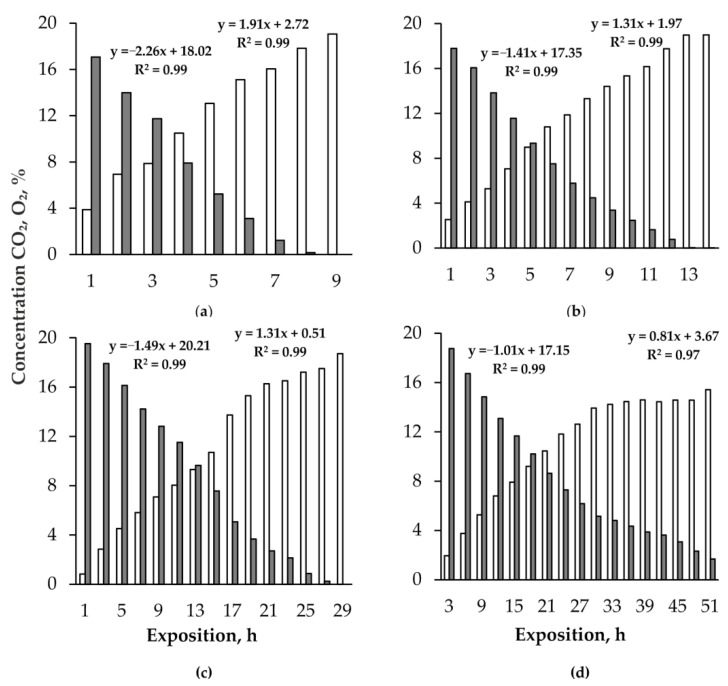
Dynamics of O_2_ consumption and CO_2_ accumulation in exposure chambers with basidiocarps *Fomes fomentarius* (**a**), *Fomitopsis pinicola* (**b**) and substrates *Fomes fomentarius* (**c**), *Fomitopsis betulina* (**d**). Dark columns—O_2_, light columns—CO_2_, R^2^—coefficient of determination.

**Figure 3 jof-08-01296-f003:**
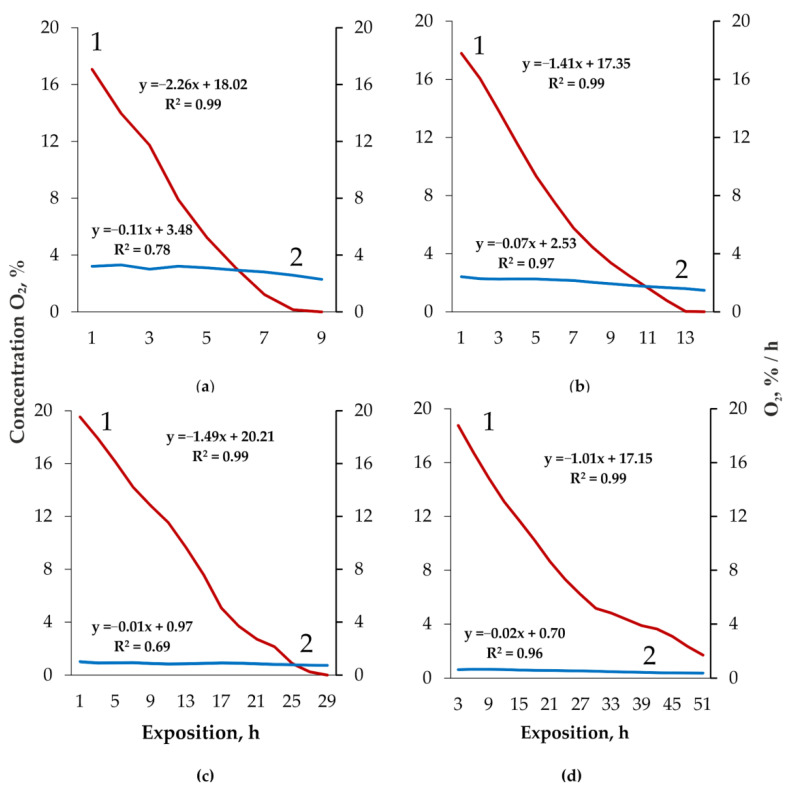
O_2_ concentration (1) and intensity of its consumption (2) by *Fomes fomentarius* (**a**), *Fomitopsis pinicola* (**b**) basidiocarps and *Fomes fomentarius* (**c**), *Fomitopsis betulina* (**d**) substrates. R^2^—coefficient of determination.

**Figure 4 jof-08-01296-f004:**
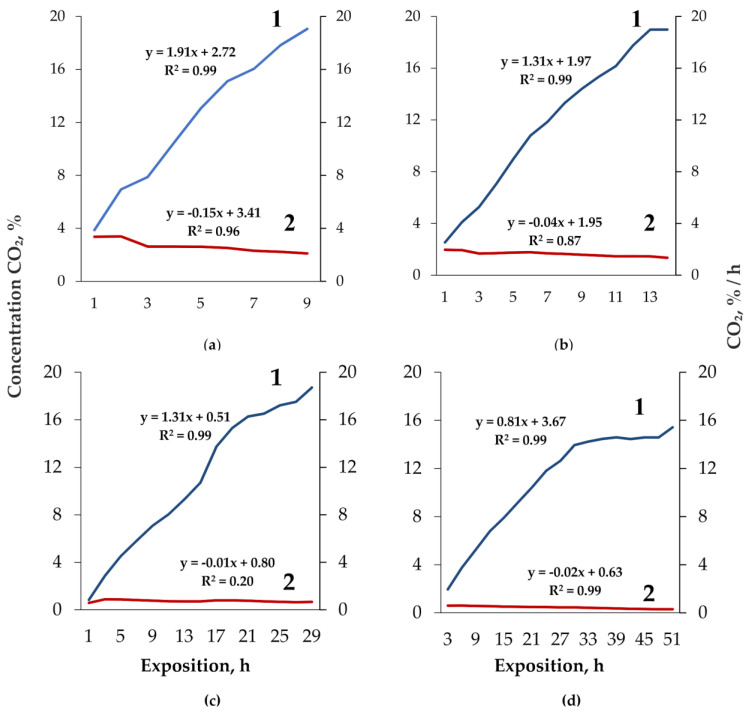
CO_2_ concentration (1) and intensity of its emission (2) by *Fomes fomentarius* (**a**), *Fomitopsis pinicola* (**b**) basidiocarps and *Fomes fomentarius* (**c**), *Fomitopsis betulina* (**d**) substrates. R^2^—coefficient of determination.

**Figure 5 jof-08-01296-f005:**
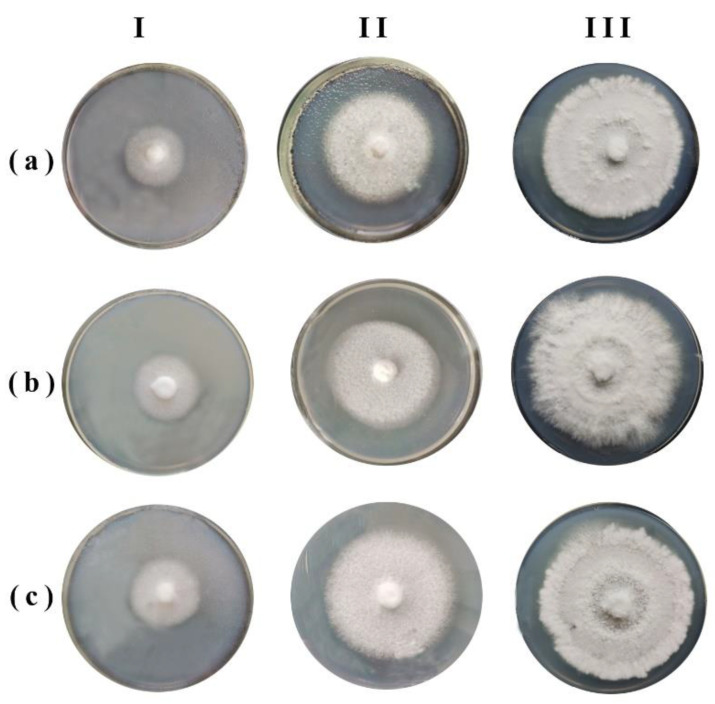
Growth of dikaryotic mycelium of *Daedaleopsis septentrionalis* in 100% CO_2_ (**a**) and in 100% N_2_ (**b**). I—initial size, II—after 3 days exposure in oxygen-free medium, III—after the next 3 days in air; control in air (**c**), initial size (I), after 3 (II) and 6 (III) days in air.

**Table 1 jof-08-01296-t001:** Ratio CO_2_ emission and O_2_ consumption by basidiocarps and substrates at 5/10/20% O_2_ in nitrogen.

Species	Substrates	Basidiocarps
*Daedaleopsis tricolor*	07/0.7/0.9	0.8/0.8/0.8
*Fomes fomentarius*	1.1/0.7/0.9	0.9/0.8/0.9
*Fomitopsis betulina*	0.9/0.7/0.9	1.0/1.1/0.9

**Table 2 jof-08-01296-t002:** Emission CO_2_ (vol. %/h) by basidiocarps and substrates at 5/10/20% O_2_ in nitrogen.

Species	Substrates	Basidiocarps
*Daedaleopsis tricolor*	0.4/0.3/0.3	0.1/0.2/0.2
*Fomes fomentarius*	1.2/1.2/1.2	2.5/2.7/2.5
*Fomitopsis betulina*	0.4/0.2/0.4	1.4/1.5/1.3

**Table 3 jof-08-01296-t003:** Ratio CO_2_ emission and O_2_ consumption by basidiocarps and substrates at 0.04/5/10/20% CO_2_ in the air.

Species	Substrates	Basidiocarps
*Daedaleopsis tricolor*	0.8/07/0.8/2.0	0.8/0.8/0.8/3.8
*Fomes fomentarius*	0.9/0.7/1.1/2.3	0.8/0.8/0.7/3.1
*Fomitopsis betulina*	0.8/1.0/0.5/2.2	0.8/0.9/0.6/1.2

**Table 4 jof-08-01296-t004:** O_2_ consumption (vol. %/h) by basidiocarps and substrates at 0.04/5/10/20% CO_2_ in the air.

Species	Substrates	Basidiocarps
*Daedaleopsis tricolor*	0.7/1.3/1.4/0.6	0.6/0.4/0.6/0.1
*Fomes fomentarius*	3.4/3.0/3.7/0.9	2.2/2.8 /1.9/1.3
*Fomitopsis betulina*	0.7/0.6/1.2/0.2	0.4/ 0.8/0.8/0.3

**Table 5 jof-08-01296-t005:** O_2_ consumption (vol. %/h) by *Daedaleopsis tricolor* basidiocarps and substrates in air before and after 72 h exposure in chambers with 100% CO_2_ concentration, *n* = 3.

Subject of Study	Gas Exchange	
	Before Exposure	After Exposure
Substrates	0.59 ± 0.07	0.46 ± 0.06 (*p* = 0.19)
Basidiocarps	0.88 ± 0.01	0.79 ± 0.03 (*p* = 0.07)

Note: *p*—the test for significance of differences (*ANOVA*).

**Table 6 jof-08-01296-t006:** Growth rate (mm/day) of dikaryotic mycelium *Daedaleopsis septentrionalis* on wort agar in oxygen-free and air media, n = 3.

Gaseous Medium	3 Days in Oxygen-Free Medium	3 Days in Air, after Oxygen-Free Medium
CO_2_	3.58 ± 0.51 (*p* = 0.01) *	4.25 ± 1.32 (*p* = 0.66) **
N_2_	7.31 ± 0.50 (*p* = 0.26) *	5.59 ± 0.55 (*p* = 0.08) **
Air (control)	8.57 ± 0.88	6.2 ± 0.84 (*p* = 0.19) **

Note: significance of differences *—with control, **—with the previous period.

**Table 7 jof-08-01296-t007:** Emission CO_2_ (vol. %/h) by basidiocarps and substrates in the air and in oxygen-free nitrogen medium.

Species	Before Exposure in Air	Exposure in Oxygen-Free Medium, h		
		16	21	45
*Fomes fomentarius*	0.26/0.78 *	0.21/0.23	0.21/0.20	0.22/0.22
*Hapalopilus rutilans*	1.39/0.29 *	0.35/0.15	0.31/0.08	0.22/0.09
*Trichaptum biforme*	1.04 **	0.26	0.27	0.23
*Fomitopsis betulina*	0.18 **	0.11	0.17	0.13

Note: *—basidiocarps/substrates; **—substrates.

**Table 8 jof-08-01296-t008:** O_2_ consumption (vol. %/h) by basidiocarps and substrates in the air before and after three-day exposure in oxygen-free nitrogen medium, n = 4.

Species	Subject of Study	Gas Exchange	
		Before Exposure in Air	After Exposure in Oxygen-Free Medium
*Daedaleopsis tricolor*	substrate	0.40 ± 0.01	0.24 ± 0.01 (*p* = 0.001) *
*D. tricolor*	basidiocarp	0.25 ± 0.01	0.12 ± 0.01 (*p* = 0.001)
*Fomitopsis betulina*	substrat	0.87 ± 0.10	0.68 ± 0.04 (*p* = 0.13)
*F. betulina*	basidiocarp	0.59 ± 0.05	0.39 ± 0.01 (*p* = 0.01)
*Fomes fomentarius*	substrat	1.05 ± 0.02	0.86 ± 0.01 (*p* = 0.001)
*F. fomentarius*	basidiocarp	1.11 ± 0.06	1.11 ± 0.05 (*p* = 0.99)

Note: *p*—the test for significance of differences (*ANOVA*).

## Data Availability

Not applicable.

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
