# Peer review of "Eco-Physiological Adaptations of the Xylotrophic Basidiomycetes Fungi to CO2 and O2 Mode in the Woody Habitat"

_jof, 2022, doi:10.3390/jof8121296_

Round 1
Reviewer 1 Report
This is an interesting and well designed research. The authors proposed a hypothesis and proved it by experiments and statistics data.
I have two small suggestions:
1. the title of the paper seems too broad, and can't not reflect the contents of the research;
2. please write some words in the abstract to indicate the aims of this research.
Author Response
We are deeply grateful to the reviewer for the positive assessment of our work! In the review there are 2 small suggestions.
1. The title of the paper seems too broad, and can't reflect the contents of the research.
We accept this proposal and give a new title for the article, which, in our opinion, fully reflects it content.
2. Please write some words in the abstract to indicate the aims of this research.
We accept this proposal and the purpose of the study in the abstract have been indicated.
Please see the attachment.

Reviewer 2 Report
Dear colleagues, the work seems interesting for me. I have two questions:
Xylotrophic basidiomycetes are the most active degraders of lignin in nature. The complex of enzymes that catalyze the degradation of lignin (laccase, peroxidase) is oxygen-dependent. How does this compare with the data you received?
The growth of fungi is more correctly assessed by the content of ergosterol (not by radial growth). This is the generally accepted technique.
Author Response
We are deeply grateful to the reviewer for the positive assessment of our work!
There are 2 questions that the esteemed reviewer would like to have answered.
1. Xylotrophic basidiomycetes are the most active degraders of lignin in nature. The complex of enzymes that catalyze the degradation of lignin (laccase, peroxidase) is oxygen-dependent. How does this compare with the data you received?
This issue is not specifically considered in the article, but our data show that brown rot fungi (Fomitopsis betulina, F. pinicola) have a significantly (2-4 times) lower intensity of oxygen consumption compared to white rot fungi (Fomes fomentarius). Perhaps this is due to the need for additional volumes of oxygen for the decomposition of lignin by rot white fungi. More detailed studies of the features of oxygen gas exchange in brown and white rot fungi we are planned for 2023.
2. The growth of fungi is more correctly assessed by the content of ergosterol (not by radial growth). This is the generally accepted technique.
In our work, studies on the growth of mycelium of fungi occupy a small, but very important part. We think it sufficient to solve the questions of interest to us, to evaluate the development of the mycelium in terms of its linear dimensions, since we had to show that the mycelium grows in an anoxic environment. Traditional methods make it possible to show visually (in the revised version of the manuscript, photos of mycelium growing on agar in an oxygen-free environment are given!) and give a comparative assessment of the growth rate of mycelium depending on the composition of the gaseous medium: N2 or CO2.

Reviewer 3 Report
Comments 1. Xylotrophic fungi species should be mentioned in the abstract 2. Wood species should be mentioned in the abstract, and why you selected those species. 3. How did you identify the xylotrophic fungi. 4. R2 values in Figures 1,2, and 3? 5. The article lacks some important phots in both materials and methods and results.
Author Response
We are deeply grateful to the reviewer for the positive assessment of our work! In the review there are 5 comments.
1. Xylotrophic fungi species should be mentioned in the abstract.
We accept this and in the abstract all 7 species studied fungi are indicated.
2. Wood species should be mentioned in the abstract, and why you selected those species.
We accept this and in the annotation 2 forest-forming tree species in the study area are named.
3. How did you identify the xylotrophic fungi?
In the Materials and Methods section of the article, appropriate changes were made, which noted that the diagnosis of fungi was carried out using traditional mycological methods based on the anatomical and morphological features of basidiocarps.
4. R2 values in Figures 1, 2, and 3?
We have added the R2 values in figures 1, 2 and 3.
5. The article lacks some important photos in both materials and methods and results.
9 photos showing the results of exposure of dikaryotic mycelium under anoxic conditions in nitrogen, carbon dioxide and in air were prepared and included in draft of the paper.
In the reviewer contains a wish to critically check the list of cited works for their relevance to the article. We did this and excluded three works from the list: two of them were published in the first half of the last century and one work in Russian. The rest of the work, in our opinion, is relevant.

Round 2
Reviewer 3 Report
The authors replied to my comment, but even if they didn't supply the molecular identification of the studied fundi, they request to insert the photos of morphological identification.
Author Response
Dear reviewer,
once again, thank you for your positive assessment of our work. We prepared photos of fungi and substrates, however in a dry state.
